# Correlates of Physical Activity of Children and Adolescents with Autism Spectrum Disorder in Low- and Middle-Income Countries: A Systematic Review of Cross-Sectional Studies

**DOI:** 10.3390/ijerph192316301

**Published:** 2022-12-05

**Authors:** Tianwei Zhong, Hui Liu, Yan Li, Jing Qi

**Affiliations:** College of Physical Education and Health Sciences, Zhejiang Normal University, Jinhua 321004, China

**Keywords:** children and adolescents, disabilities, physical activity, socio-ecological model, review

## Abstract

Children and adolescents with autism spectrum disorder (ASD) are at a high risk for a lack of physical activity (PA). The aim of this study is to review the evidence on the correlates of PA in children and adolescents with ASD in low- and middle-income countries. We searched Psychology and Behavioral Sciences Collection (PBSC), Scopus, PsycINFO, Web of Science (WOS), MEDLINE, Education Resources Information Center (ERIC), Education Source (ES), and Academic Search Premier (ASP) databases for relevant studies until April 2022, inclusive, to examine the factors associated with the studies of PA in children and adolescents with ASD aged 5 to 17 years in low- and middle-income countries. A total of 15 articles are included in the present review. Three researchers assessed the methodological quality and extracted relevant data of the included reviews. The correlates were synthesized and further assessed semi-quantitatively. Results of this review show that gender (boys) and more PA opportunities were positively associated with the PA of children and adolescents with ASD, while age and body mass index (BMI) were negatively related to their PA levels in low- and middle-income countries. The day of week was found to be inconsistently associated with PA in children and adolescents with ASD. The findings suggest that research on the correlates of PA in adolescents with ASD in low- and middle-income countries is limited. However, there are clear correlates for which future interventions could be based (age, gender, BMI, and PA opportunity) to promote PA participation in children and adolescents with ASD in low- and middle-income countries.

## 1. Introduction

Autism spectrum disorder (ASD) is clinically defined by impairments in social communication and reciprocal interaction, and shows repetitive, restricted, and stereotypical behavioral patterns [1]. The intelligence of individuals with ASD greatly differs and most of them need lifelong care from family and society, which greatly influences patients’ physical and mental health and socioeconomic development. Meanwhile, accumulating evidence has indicated that ASD children and adolescents are at a higher risk of obesity than the general population [1,2,3,4]. ASD has been emerging as a major public health problem that has spread beyond geographic, economic, and social boundaries [5]. ASD begins before the age of three and persists throughout a person’s life, and it has an average prevalence of one in 150 children and adolescents worldwide [6]. A proportion of 90% of those with ASD live in low- and middle-income countries, such as in Africa, South and West Asia, and South America [7,8]. In addition, females with ASD suffer more biases and are more prone to abuse than males with ASD in low- and middle-income countries [9]. For example, girls in Nepal have reduced access to health facilities compared to boys due to a high preference for boys [10]. More investment in ASD research in low- and middle-income countries is needed given the rapid spread of awareness about ASD around the world.

Physical activity (PA) is defined as any bodily movement produced by skeletal muscles that result in energy expenditure [11]. Regular participation in PA has physical, psychological, and social benefits for children and adolescents with disabilities [12,13,14], including improving health and physical fitness [15]; maintaining normal muscle strength, flexibility, and function [16,17]; increasing self-confidence and self-efficacy [18]; and building friendships and enhancing social skills [19]. Thus, the WHO recommends that children and adolescents with disabilities should engage in at least 60 min of moderate-to-vigorous-intensity PA (MVPA) per day [20].

Engaging in sufficient PA is the key promoting factor of health for children and adolescents with ASD. However, significant impairments in motor, social communication, sensory, and behavioral domains may limit the opportunities for engagement in PA among this population [21,22,23,24]. For example, social communication impairments of children and adolescents with ASD may limit their abilities to participate in group sports. Moreover, behavioral problems or preferences among this population (e.g., restricted interests, inflexible schedules, and preference for predictable, structured activities) may limit their PA choices (e.g., playing games, transportation, and engaging in chores, recreation, and other unstructured activities) [24,25]. Numerous studies have demonstrated that children and adolescents with ASD across countries (e.g., United States and Canada) do not meet the international PA recommendations [26,27,28,29]. For example, McCoy and Morgan examined the PA levels of 1036 American children and adolescents with ASD aged 10–17 and found that only 11% of adolescents with ASD met the international PA guidelines [28]. The results of Bremer et al. also showed that only 14.9% of Canadian children and adolescents with ASD aged 6–13 years met the international PA guidelines in their study [27]. Previous studies also proved that the PA levels of children and adolescents with ASD were lower than those of their peers without disabilities [30,31,32].

Many studies have examined the influencing factors of PA participation among children and adolescents with ASD using self-reporting and objective measures (e.g., accelerometers). Gender, age, body mass index, ASD severity level, and day of the week (week versus weekend day) were examined, and the results of previous studies were inconsistent [22,33]. For example, the results of Memari et al. showed that gender was a significant influencing factor of PA of children and adolescents with ASD [23], while Macdonald et al. found no significant difference between male and female children and adolescents with ASD [34]. The results of Pan et al. also showed that age was a significant influencing factor of PA of children and adolescents with ASD [24], while Gehricke et al. found no significant effect of age on PA in children and adolescents with ASD [35]. The inconsistences among studies suggest the need for further investigation of the influencing factors of PA levels in children and adolescents with ASD.

To date, four reviews have analyzed the influencing factors of PA participation in children and adolescents with ASD [22,33,36,37]. Two previous reviews have analyzed the influencing factors of PA of children and adolescents with ASD [22,33]. Jones et al. reviewed 10 PA-relevant studies published before November 2015 and concluded that the PA of children and adolescents with ASD was negatively associated with their age, school levels, and grades [22]. On the basis of the socio-ecological model, Arkesteyn et al. reviewed 32 studies published before March 2021 and made a distinction between PA correlated in children and adolescents. The review concluded that three of the four levels of the socio-ecological model (i.e., the intrapersonal, social, and environmental levels) influenced the PA levels among children with ASD, and only intrapersonal and social factors influenced the PA levels among adolescents with ASD [33]. The two reviews conducted an overview of the influencing factors of PA levels of children and adolescents with ASD, which did not differentiate between participants from high-, low-, and middle-income countries. The low- and middle-income countries have the most substantial proportion of children with ASD but with the most minimal research and services available [38]. The results of previous studies have shown that children and adolescents with ASD from low- and middle-income countries do not meet the international PA guidelines [31,39,40], and their PA levels were lower than those of their peers without disabilities [40,41,42,43]. For example, only 44% of children and adolescents with ASD in China met the international PA guidelines [39]. The research findings on PA level comparisons show that the daily PA of children and adolescents with ASD is significantly lower than their peers without disabilities in China [43]. Moreover, some studies have examined the factors influencing the PA of children and adolescents with ASD in low- and middle-income countries. For examine, Pan et al. (2021) concluded that age was a big factor influencing PA levels of children and adolescents with ASD in low- and middle-income countries [24]. However, no reviews to our knowledge examined the influencing factors of PA levels of children and adolescents with ASD from low- and middle-income countries. Therefore, this review aims to systematically summarize the studies examining the correlates of PA of children and adolescents with ASD in low- and middle-income countries. We focused on potential correlates at all five variables of the socio-ecological model (i.e., correlates at the demographic and biological variables; psychological, cognitive, and emotional variables; behavioral attributes and skills; social and cultural variables; and physical environment variables [44,45,46]) for children and adolescents with ASD in low- and middle-income countries. The findings will aid in developing effective interventions to improve the PA in children and adolescents with ASD in low- and middle-income countries and propose directions for future research.

## 2. Methods

This study was based on the guidelines of the Preferred Reporting Items for Systematic Reviews and Meta-Analyses (PRISMA) to identify, screen, and evaluate qualifications and decide to include and exclude studies [47]. It was registered on PROSPERO (CRD42022329113).

### 2.1. Search Strategy

Psychology and Behavioral Sciences Collection (PBSC), Scopus, PsycINFO, Web of Science (WOS), MEDLINE, Education Resources Information Center (ERIC), Education Source (ES), and Academic Search Premier (ASP) databases were searched for relevant studies. The search strategies included keywords in four categories, namely, (1) “physical activit*” OR “PA” OR “MVPA” OR exercise* OR “motor activit*” OR “sport*” OR fitness* OR “physical education”; (2) youth* OR “young athlete*” OR adolescent* OR teenager* OR child* OR childhood* OR student*; (3) “autism*” OR “autistic*” OR “autism spectrum disorder*” OR “ASD” OR “asperger syndrome” OR “pervasive developmental disorder” OR “PDD”; and (4) correlate* OR factor* OR reason* OR predictor*. A manual search through the reference lists of included studies was also conducted to identify other potentially relevant studies.

### 2.2. Inclusion and Exclusion Criteria

The inclusion and exclusion criteria were as follows: (1) cross-sectional studies which examined the associations between PA and other variables were covered; case-control, prospective, retrospective cohort, case studies, qualitative studies, expert opinion, or experimental studies where a particular correlate could not be extracted were excluded; (2) studies that explored the potential associations between PA as a quantitatively measured outcome variable and independent variables not as part of an intervention; (3) studies that targeted children or adolescents with ASD (aged 5 to 17 years or have a mean age in this range) were included, whereas those beyond that age group who did not have ASD were excluded; (4) studies with participants from low- and middle-income countries as defined by the World Bank (i.e., low-income countries with a gross national income per capita of $1005 or less; middle-income countries had a gross national income per capita of $1006–$12,235) [8] were covered, whereas research with participants not from low- and middle-income countries were excluded; and (5) studies published in peer-reviewed journals and with full-text articles in English from database inception to April 2022 were included; on the contrary, unpublished articles, comments, conference proceedings, and dissertations were excluded. The identified studies were determined by three reviewers (T.W.Z., H.L., and Y.L.) independently according to the inclusion and exclusion criteria. A fourth reviewer (J.Q.) was consulted in case of disagreement. The final number of studies identified in this review was 16. Figure 1 summarizes the process of literature selection.

### 2.3. Quality Assessment

This study used the revised McMaster Quantitative Critical Review Forms (Supplement 1) [48,49] to evaluate the methodological quality of the included studies. The form was selected because it shows a good consistency of 75–86% [50] among evaluators and has been used to assess the methodological quality of research in similar areas [51,52]. The Critical Review Form consists of 16 items: study purpose (1 item), background (1 item), design (1 item), sampling (2 items), measurement (4 items), data analysis (4 items), conclusions (1 item), and implications and limitations (2 items). All items were scored by the degree to which specific criteria were met (yes = 1, no = 0, not applicable = NA). The researchers calculated the summary score for each study by summing the total score obtained across relevant items and dividing it by the total possible score. The scores of ≤50%, 51–75%, and >75% were interpreted as low, good, and excellent quality for individual studies, respectively [49]. The first and second authors independently performed the methodological quality assessment. If consensus could not be reached, then the agreement was obtained through discussion with the third author.

### 2.4. Data Extraction and Analysis

Data were extracted and presented according to the characteristics of the included studies. Extract information includes the geographic location, sampling strategy, participant details (i.e., sample size, age, gender, and ASD level), PA measurement methods, dependent variable, and analytical approaches.

According to the socio-ecological model, the factors correlated with PA are categorized as follows: (a) demographic and biological variables (e.g., gender, age); (b) psychological, cognitive, and emotional variables (e.g., self-esteem, perceived benefits); (c) behavioral attributes and skills (e.g., healthy diet, sedentary pursuits); (d) social and cultural variables (e.g., parental PA, parent benefits of PA); and (e) physical environment variables (e.g., season, time outdoors) [44,45,46]. Variables were classified as “related” or “not related” to PA, which was based on statistical significance (*p* < 0.05). The pooled associations between potential correlates and PA were classified as: (a) positive associations (denoted by ‘+’); (b) negative associations (denoted by ‘−’); or (c) nonsignificant or inconsistent associations, which were indicated in the column “not related to PA”. A summary code for each potential correlate was given using previous recommendations [46,53]. The summary code column contains a code to summarize the literature for that specific correlate. According to Sallis et al. [46], variables that were compared less than three times were not described in the main text. However, in this review, these variables are included in the tables to understand the variable screening procedures. This review used a semi-quantitative evaluation method to determine the consistency of the different types of potentially relevant factors. This semi-quantitative procedure provides additional objective evidence rather than only a report of narrative results. Variables reported in fewer than three studies were coded as “no description (ND).” For variables that occurred more than three times, the direction of association was based on the rules developed by Sallis et al. [46]: associations with 0–33% of similar directions were considered unrelated and coded as “0”; associations with 34–59% of similar directions were defined as uncertain or inconsistent and coded as “?”; and 60–100% of associations in similar directions were considered consistent and coded as “++” (positive) or “−−” (negative). The percentages refer to the number of significant associations with the variable divided by the total number of times the variable was studied in the literature. This cutoff coding has been used in previous reviews [22,54,55].

## 3. Results

### 3.1. Search Results

The initial search yielded 1424 studies. After duplicates from the original sample were removed (*n* = 389), the remaining 1035 articles were screened based on the title and abstract, and 971 studies were excluded because they were not relevant to the current research topic, nonoriginal in nature, not peer-reviewed journals, or not a full-text article. The remaining 64 full-text articles were read and another 49 were rejected. The main reasons for exclusion were that the results were not applicable to the type of study (*n* = 11), the participants were not in developing countries (*n* = 23), the participants did not have ASD (*n* = 1), the focus was not on the PA correlates (*n* = 13), and the age or mean age of the participants were out of the 5–17 years range (*n* = 1). Therefore, 15 studies were included in this review. Figure 1, adapted from the PRISMA group [47], displays the detailed search and study selection process.

### 3.2. Methodological Quality

The results of the assessment of the methodological quality are outlined in Appendix A. All studies included in this review displayed clear research purposes, had relevant background literature, used appropriate research designs, obtained informed consent, reported results in terms of statistical significance, adopted appropriate analysis methods, reported the importance of the practice, and drew an appropriate conclusion. Over 80% of the studies reported dropouts, acknowledging and describing study limitations, and described research methods in detail and contributed to future practice. Valid and reliable tools were used in 11 (73.3%) and 12 (80%) studies, respectively. The sampling part was poorly reported with only four (26.7%) studies that had sample sizes justified and 11 (73.3%) studies described the sample in detail. In total, twelve studies (80%) were categorized as ‘excellent’ and three (17.8%) were considered ‘good’.

### 3.3. Description of Studies

Table 1 summarizes the studies by the first author, geographic location, sampling strategy, participant details, and PA measure. Of the 15 included studies, 11 studies (73.3%) were conducted in Chinese Taiwan, 3 studies (20%) in Tehran, Iran, and 1 study (6.7%) in Istanbul, Turkey. Two studies (13.3%) adopted questionnaires to measure PA. Objective measures were adopted in 12 studies (80%), such as accelerometer, and one study (6.7%) adopted the two approaches. Of these 15 studies, four studies did not indicate the sampling strategy. The remaining studies used purposive sampling (*n* = 4) and convenience sampling (*n* = 7). Sample sizes ranged from 25 to 215, 1 study had a sample size above 100 and 5 studies had less than 50 participants. Eleven studies (73.3%) used univariate analysis to evaluate the association between variables and the PA of children and adolescents with ASD in low- and middle-income countries, 1 study (6.7%) reported multivariate analysis results, and 3 studies (20%) adopted the two approaches.

### 3.4. Correlates of PA of Children and Adolescents with ASD

Table 2 provides a summary of potential PA correlates in children with ASD in low- and middle-income countries. A total of 25 factors were identified in the review, with five (20%) factors assessed three or more times. Of these, four variables were associated with the PA of children and adolescents with ASD in low- and middle-income countries, and one variable was found to be inconsistently associated with PA.

#### 3.4.1. Demographic and Biological Variables

Nine variables (36%) were categorized as demographic and biological variables related to the PA of children and adolescents with ASD in low- and middle-income countries. Demographic variables included age, gender, body mass index (BMI)/obesity, severity of disorder, medication use, comorbidities, physical fitness, socioeconomic status, and social functioning. Of these, age, gender, and BMI were studied three or more times. Among the three variables, age was the most frequently studied correlate (eight studies) that was found to be negatively associated with the PA of children and adolescents with ASD in low- and middle-income; the result was due to the fact that 87.5% of the associations were in the same direction [23,24,56,57,58,60,61,62,64]. Gender was another main demographic-influencing factor of PA of children and adolescents with ASD in low- and middle-income countries. The results of four studies reported that boys were significantly more active than girls [23,57,61,62]. Three studies showed evidence for a negative association between BMI and PA of children and adolescents with ASD in low- and middle-income countries [40,61,63]. The other variables were studied less than three times. Therefore, these variables were not described and discussed.

#### 3.4.2. Psychological, Cognitive, and Emotional Variables

Two variables (e.g., motivation and cognitive flexibility, 8%) [23,42] were categorized as psychological, cognitive, and emotional variables related to the PA of children and adolescents with ASD in low- and middle-income countries. They were not described and discussed because they appeared in less than three comparisons.

#### 3.4.3. Behavioral Attributes and Skills

Sedentary pursuits [56,61] were categorized as behavioral attributes and skills related to the PA of children and adolescents with ASD in low- and middle-income countries, which were not described and discussed because they appeared in less than three comparisons.

#### 3.4.4. Social and Cultural Variables

Eight variables (32%) were identified from the reviewed studies as social and cultural variables related to the PA of children and adolescents with ASD in low- and middle-income countries, namely, parent PA [56], parental support [56], types of PA programs and facilities [57], social support and instruction [41,58], social engagement [59], household structure [62], parental education level [23,62], and PA opportunities [43,60,62]. In this group, all variables except for the PA opportunities were studied less than three times. Therefore, these variables were not described and discussed. Meanwhile, PA opportunity was positively associated with the PA of children and adolescents with ASD in low- and middle-income countries, which was due to the fact that 100% of the associations were in the same direction [43,60,62].

#### 3.4.5. Physical Environment Variables

Six variables (24%) were identified from the reviewed studies as physical environment variables related to the PA of children and adolescents with ASD in low- and middle-income countries, namely, day of week (weekdays versus weekend days) [24,57,60,61], PE class-related factors [31,59], during school time [57], recess time [41,57], and school environment [31]. Of these variables, only day of week (weekdays versus weekend days) was studied three or more times. Thus, other variables were not described and discussed. Day of week (weekdays versus weekend days) was found to be related to the PA of children and adolescents with ASD in low- and middle-income countries in 42.9% of the comparisons [24,40,57,61]. Therefore, the relationship was inconclusive.

## 4. Discussion

For this review, we identified potential variables correlating to the PA of children and adolescents with ASD in low- and middle-income countries from 15 studies. The results of the methodological quality assessment showed that all the studies reviewed were identified as having medium and high quality. Therefore, the findings of this review can be used to draw a meaningful conclusion on the influencing factors of the PA of children and adolescents with ASD in low- and middle-income countries.

Among the 25 variables assessed from the reviewed studies, only five variables were reported three or more times. Four variables were categorized as having “consistent association” with the PA of children and adolescents with ASD in low- and middle-income countries. Among them, certain correlates, such as gender (boys) and more PA opportunities, were positively associated with the PA of children and adolescents with ASD. The other variables, including age and BMI, were negatively related to the PA of children and adolescents with ASD. The remaining one variable (day of week) was placed in the “inconsistent” category. The results of the review are partially consistent with those of Arkesteyn et al. [33].

Similar to Arkesteyn et al. [33], our review also concluded that age was negatively related to the PA of children and adolescents with ASD. Studies have found that primary school children with ASD are more active (i.e., performed more MVPA) than secondary school adolescents with ASD [24,57]. Several factors may make children and adolescents with ASD more sedentary as they increase with age. First, increasing homework or participation in organized activities during the afterschool periods from children to adolescents is likely responsible for decreases in PA [57,65,66]. Moreover, children and adolescents with ASD may have increased sedentary and screen time with age, which may limit their PA participation [56,61,67]. Finally, as children with ASD grow older, the social deficits experienced by children with ASD to adolescents with ASD may result in a reluctance to engage in play and sports that are often socially demanding. This aversion to group games may lead to a preference for individual activities such as television viewing and video games, especially nonsocial screen-based activities [68]. Transition planning for children and adolescents with ASD to facilitate active PA participation as they grow is warranted [61]. Future research should carefully investigate the reasons for age-related declines in PA among children and adolescents with ASD in low- and middle-income countries.

Our review concluded that a significant difference existed between genders on the PA engagement of children and adolescents with ASD in low- and middle-income countries. This finding is inconsistent with that of Arkesteyn et al. [33], who found that genders were not associated with PA in children with ASD. One of the main reasons for this inconsistency may be that the three studies included in this review conducted in Iran found that boys had significantly higher PA levels than girls in their countries [23,60,62]. In Iran, negative attitudes are generally held towards females’ PA and other sports activity engagements. These distinct cultural barriers may result in limited PA involvement of females [69,70,71]. However, this result needs to be interpreted with caution because it represents Iran only rather than low- and middle-income countries. Future studies are suggested to continue to examine the gender difference in PA among children and adolescents with ASD.

BMI/obesity was found to be negatively related to the PA of children and adolescents with ASD in low- and middle-income countries. This result is inconsistent with the finding of Arkesteyn et al., who did not conclude on the relationship of BMI/obesity with the PA of children and adolescents with ASD [33]. Lack of PA among obese children and adolescents with ASD in low- and middle-income countries may mean excessive sedentary behavior [61]. Salvy and Bowker proposed that the peer social context of children and obesity of adolescents, characterized by social stigma and peer difficulties, contributed to and reinforced the lack of PA and choice of sedentary alternatives of overweight/obese children and adolescents [72]. Children and adolescents who are obese or overweight are more susceptible to isolation and bullying by normal weight children [4], which may result in their negative attitudes toward engaging in PA, which in turn reduces PA levels. In low- and middle-income countries, overweight or obese children and adolescents may be more likely to be bullied owing to their physical appearance. Addressing bullying victimization in low- and middle-income countries can potentially reduce the risk for weight gain and obesity among adolescents via the reduction in obesogenic behaviors in this setting [73]. Therefore, practitioners, teachers, and parents of children and adolescents with ASD in low- and middle-income countries need to prioritize practicing PA and keeping their BMI in the normal range. School-based approaches to developing anti-bullying programs are also recommended, including parent meetings, firm disciplinary methods, and improved playground supervision [74].

PA opportunities were found to be positively related to the PA participation of children and adolescents with ASD in low- and middle-income countries. Lack of PA opportunities could limit PA participation of children and adolescents with ASD, and the lack of PA opportunities exists within and outside of school settings [43,60,62]. In Taiwan, many regular secondary schools have eliminated PE and recess to increase academic opportunities for their students [43]. Previous research has suggested that students with ASD typically accumulated the most MVPA during PE [41] and recess [56]. Removing these PA periods of school time would deprive many ASD students of crucial PA opportunities. In Iran, Memari et al. [61] also found that reduced PA opportunities in autism-specific schools contributed to a reduction in the total individual PA for children and adolescents with ASD. Studies in Taiwan and Iran have also found that a lack of opportunities to extracurricular PA may also be a reason for reduced PA participation in children and adolescents with ASD. These studies found a lack of integrated and segregated extracurricular PA programs for children and adolescents with ASD [43,62]. Therefore, schools in low- and middle-income countries need to ensure that children and adolescents with ASD have adequate PE classes and recess during the school day. They should also provide environmental support and teacher guidance for their participation in afterschool PA to increase PA opportunities for children and adolescents with ASD. PA participation among children and adolescents with ASD should be promoted as well in low- and middle-income countries.

Novelty of this review.

The strengths of this study include the extensive and systematical searches that were conducted in multiple databases to identify the related literature. This systematic review is also the first to use a semi-quantitative evaluation to identify the correlates of the PA of children and adolescents with ASD in low- and middle-income countries. This evaluation was conducted to obtain more information on the consistency of the reported associations by integrating the results from all relevant studies regardless of whether the association was statistically significant.

Limitations and Implications of this review.

The present review has several limitations. The first limitation is that only English -language-published studies were included in our review. Papers published in Chinese might have given more information on this topic. Second, the present review was similar in most aspects to previous reviews (i.e., age and day of week). Third, a semi-quantitative evaluation was adopted to analyze the research findings of the studies included. This approach addressed the significance and direction of each association to determine the consistency of the reported association, and could not assess the strength or magnitude of these associations. Future studies are expected to investigate the interactions of these correlates across various domains. Fourth, 11 of the 15 included studies (73.3%) were conducted in Taiwan, China, and they represent the correlation between PA levels and ASD in children and adolescents in Taiwanese Chinese culture. Further studies from other low- and middle-income countries are needed to identify any cultural differences.

## 5. Conclusions

The purpose of this study is to examine the influencing factors of PA levels of children and adolescents with ASD in low- and middle-income countries. The present review demonstrates that gender (boys) and PA opportunities are positively associated with the PA of children and adolescents with ASD, and that age and BMI are negatively associated with the PA of children and adolescents with ASD in low- and middle-income countries. The current relationship between day of the week (weekdays versus weekend days) PA participation and PA participation among children and adolescents with ASD in low-income countries was inconsistent and needs further investigation and determination.

The priority for the variables classified as consistently associated with PA should be to apply these findings to improve interventions. The non-modifiable demographic variables suggest that subgroups of relatively inactive young people need to be targeted for special intervention programs. Subgroups at risk for being inactive include girls with ASD, older children and adolescents with ASD, and higher BMI children and adolescents with ASD. PA opportunities identified in this review may be considered potential mediators of PA in children and adolescents with ASD, and interventions should be developed to change these variables through education, family programs, or environmental and policy changes [49]. Day of the week (weekdays versus weekend days) PA participation whose associations with PA were classified as inconsistent should be subjected to more detailed study. For example, there may be some subgroups of children and adolescents with ASD for whom day of the week (weekdays versus weekend days) PA participation is an important correlate or there may be other variables that moderate the association.

In low- and middle-income countries, the majority of studies are on the PA-related factors at the demographic and biological variables, social and cultural variables, and physical environment variables. However, there is not sufficient evidence on the psychological, cognitive, and emotional variables; behavioral attributes; and skill variables, because the majority of factors have been studied by only one study. Perhaps this suggests that research on the factors influencing PA participation among children and adolescents with ASD in low- and middle-income countries is relatively fragmented, which needs more research on the psychological variables, behavioral attributes, and skill variables. Future studies should attempt to analyze the role of multiple factors associated with PA at multiple levels, e.g., recess activities organized by teachers, creation of outdoor boy and girl zones, and regulations of electronic devices, particularly smart phones and tablets, during recess. Given that research suggests that maintaining changes in PA requires a multilevel approach [75], exploring these relationships is critical to improving our understanding of the factors influencing PA in children and adolescents with ASD.

## Figures and Tables

**Figure 1 ijerph-19-16301-f001:**
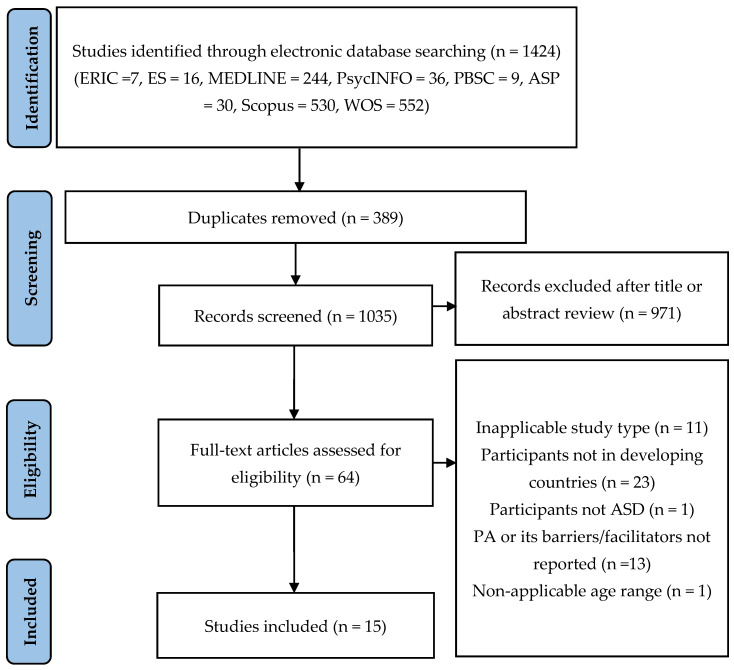
Flowchart of search and study selection.

**Table 1 ijerph-19-16301-t001:** Descriptive statistics of included studies.

First Author (Year)	Geographic Location	Sampling Strategy	Participant Details				Dependent Variables	PA Measures	Analytical Approaches
			Sample Size	Age	Gender	ASD Type			
Pan (2005) [56]	Taiwan, China	convenience	30	13.2	Male: 27 Female: 3	Autism: 14 Asperger’s syndrome: 12 PDD-NOS: 4	MVPA	accelerometer	univariate analysis multivariate analysis
Pan (2006) [57]	Taiwan, China	convenience	30	10–19	Male: 27 Female: 3	Autism: 14 Asperger’s syndrome: 12 PDD-NOS: 4	MVPA	accelerometer questionnaire	univariate analysis
Pan (2008a) [31]	Taiwan, China	convenience	24 24 (children without disabilities)	7–12	Male: 23 Female: 1	Autism (mild or high functioning: 12; moderate: 9) Asperger’s syndrome: 3	MVPA	accelerometer	univariate analysis
Pan (2008b) [41]	Taiwan, China	convenience	24 24 (children without disabilities)	7–12	Male: 23 Female: 1	Autism (mild or high functioning 12; moderate 9) Asperger’s syndrome 3	MVPA	accelerometer	univariate analysis
Pan (2009) [58]	Taiwan, China	could not be determined	25	7–12	Male: 25	Autism (mild or high functioning: 11; moderate: 8) Asperger’s syndrome (*n* = 6).	MVPA	accelerometer	multivariate analysis
Pan (2011a) [42]	Taiwan, China	could not be determined	25 75 (adolescents without disabilities)	14.08 ± 0.89	Male: 25	Mild autistic disorders: 15 Asperger’s syndrome: 10	VPA, MPA, MVPA	accelerometer	univariate analysis
Pan (2011b) [59]	Taiwan, China	could not be determined	19 76 (adolescents without disabilities)	14.19 ± 0.82	Male: 19	n/a	VPA, MPA, MVPA	accelerometer	univariate analysis
Pan (2011c) [60]	Taiwan, China	could not be determined	35	7–12	Male: 35	Mild autistic disorders: 22 Asperger’s syndrome: 13	VPA, MPA, MVPA	accelerometer	univariate analysis
Memari (2013) [61]	Tehran, Iran	purposive	90	7–14	Male: 55 Female: 35	n/a	Daily PA	accelerometer	univariate analysis multivariate analysis
Memari (2015) [62]	Tehran, Iran	purposive	83	6–15	Male: 52 Female: 31	n/a	Daily PA	questionnaire	univariate analysis
Pan (2015) [43]	Taiwan, China	convenience	30 30 (adolescents without disabilities)	12–17	Male: 30	Mild autistic disorders: 23 Asperger’s syndrome: 7	Daily PA	accelerometer	univariate analysis
Bicer (2016) [63]	Istanbul, Turkey	convenience	118 97 (adolescents without disabilities)	12–18	Male: 118	n/a	Weekly PA and Dietary Reference Intake	questionnaire	univariate analysis
Pan (2016) [40]	Taiwan, China	Convenience	35 35 (adolescents without disabilities)	12–17	Male: 30 Female: 5	Mild autistic disorders: 25 Asperger’s syndrome: 10	MVPA	accelerometer	univariate analysis
Memari (2017) [23]	Tehran, Iran	purposive	68	6–16	Male: 42 Female: 26	n/a	MVPA and sedentary time	accelerometer	univariate analysis multivariate analysis
Pan (2021) [24]	Taiwan, China	purposive	68	6–17	Male: 68	Autistic: 47 Asperger syndrome: 21	MVPA and sedentary time	accelerometer	univariate analysis

Note: ASD, autism spectrum disorder; PDD–NOS, pervasive developmental disorder–not otherwise specified; n/a, not applicable; MVPA, moderate-to-vigorous-intensity physical activity; VPA, vigorous intensity physical activity; MPA, moderate intensity physical activity; PA, physical activity.

**Table 2 ijerph-19-16301-t002:** Summary on correlates of PA of children and adolescents with ASD (*n* =15).

Correlates Variables	Related to PA		Unrelated to PA References	Summary Code	
	References	Assoc. (−/+)		Assoc.	% Studies
Demographic and Biological Variables					
Age	Pan & Frey (2005) [56]; Pan & Frey (2006) [57]; Pan et al. (2011c) [60]; Memari et al. (2013) [61]; Memari et al. (2015) [62]; Memari et al. (2017) [23]; Pan et al. (2021) [24]	−		−−	7/8 (87.5%)
	Pan (2009) [58]	+			
Gender (male)	Memari et al. (2013) [61]; Memari et al. (2015) [62]; Memari et al. (2017) [23]	+	Pan & Frey (2006) [57]	++	3/4 (75%)
BMI/obesity	Memari et al. (2013) [61]; Bicer & Alsaffar (2016) [63]	−	Pan et al. (2016) [40]	−−	2/3 (66.7%)
Severity of disorder			Memari et al. (2015) [62]	ND	
Medication use			Memari et al. (2013) [61]	ND	
Comorbidities	Memari et al. (2013) [61]	−		ND	
Physical fitness	Pan et al. (2016) [40] (Cardiovascular endurance, upper body and abdominal muscular strength)	+	Pan et al. (2016) [40] (Flexibility)	ND	
Socioeconomic status	Memari et al. (2015) [62]	−	Memari et al. (2013) [61]	ND	
Social functioning	Memari et al. (2017) [23]	+		ND	
Psychological, Cognitive, and Emotional Variables					
Motivation	Pan et al. (2011a) [42]	+		ND	
Cognitive flexibility	Memari et al. (2017) [23]	+		ND	
Behavioral Attributes and Skills					
Sedentary pursuits	Pan & Frey (2005) [56]; Memari et al. (2013) [61] (Only boy)	−		ND	
Social and Cultural Variables					
Parent PA			Pan & Frey (2005) [56]	ND	
Parental support			Pan & Frey (2005) [56]	ND	
Types of PA programs and facilities			Pan & Frey (2006) [57]	ND	
Social support and instruction	Pan (2008b) [41] (During school time); Pan (2009) [58]	+		ND	
Social engagement	Pan et al. (2011b) [59]	+		ND	
Household structure (two-parent families)	Memari et al. (2015) [62]	−		ND	
Parental education level	Memari et al. (2017) [23]	+	Memari et al. (2015) [62]	ND	
PA opportunities	Pan et al. (2011c) [60]; Memari et al. (2015) [62]; Pan et al. (2015) [43] (in school)	+		++	3/3 (100%)
Physical Environment Variables					
Day of week: weekdays (vs weekend days)	Pan et al. (2011c) [60] (upper grade); Pan et al. (2016) [40]; Pan et al. (2021) [24]	+	Pan & Frey (2006) [57]; Pan et al. (2011c) [60] (lower grade); Memari et al. (2013) [61]	?	3/7 (42.9%)
	Pan et al. (2011c) [60] (middle grade)	−			
Physical education (PE) class-related factors	Pan (2008b) [31] (PE class time); Pan et al. (20011b) [59] (PE class content)	+		ND	
During school time	Pan & Frey (2011c) [57]	+		ND	
Recess time	Pan & Frey (2011c) [57]; Pan (2008b) [41]	+		ND	
School environment	Pan (2008a) [31] (Ample spaces, equipment, and playground facilities)	+		ND	

Note: Assoc., association; the pooled associations between potential correlates and PA were classified as: positive associations (denoted by ‘+’); negative associations (denoted by ‘−‘); or nonsignificant or inconsistent associations, which were indicated in the column “not related to PA”. the percentages in parentheses refer to the number of associations supporting the expected association divided by the total number of associations for the variable. Variables were coded as “no description (ND)” when reported for fewer than three studies; 60–100% of similarly oriented associations were coded as “++” (positive) or “−−” (negative) when the variable occurred more than three times; 34–59% of similarly oriented associations were coded as “?” (inconsistent); and 0–33% of similarly oriented associations were coded as “0” (no association).

## Data Availability

The data presented in this study are available upon request from the authors.

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
