# Peer review of "Correlates of Physical Activity of Children and Adolescents with Autism Spectrum Disorder in Low- and Middle-Income Countries: A Systematic Review of Cross-Sectional Studies"

_ijerph, 2022, doi:10.3390/ijerph192316301_

Round 1
Reviewer 1 Report
The article entitled "Correlates of Physical Activity of Children and Adolescents with Autism Spectrum Disorder in Low- and Middle-Income Countries: A Systematic Review of Cross-sectional Studies" is a very interesting systematic review that clearly shows the lack of research on the subject studied. Therefore, based on the findings and conclusions drawn, future lines of research on the correlation between the practice of physical activity (PA) among children and adolescents in low-income countries can be proposed, where strategies/programs can be implemented and evaluated to promote PA practice habits in this population, in the most disadvantaged contexts, and taking into account, among other variables, PA preferences according to gender and age by intervals of children and adolescents with ASD and other disorders. Along these lines, the topic is of high and relevant interest.
However, as contributions and suggestions for improvement, the following aspects are indicated:
In the key words, it is recommended not to repeat terms similar or identical to those that already appear in the title of the work.
Despite the support of scientific literature in the Introduction section, it is suggested that the authors mention in the Introduction section what percentage of children and adolescents are affected by ASD in general and in less favoured countries. From these data, indicate the differentiated affection in boys and girls in the variables treated, as this would be interesting to deepen the gender issue that will be addressed later.
It is recommended to improve the wording of the objective by going deeper and specifying a little more in the content of what is being done. Try to provide more information. For example, cite some study factor or variables. The sentence between lines 103-105 seems more typical of the conclusions than an objective, it is suggested to change it to this section.
In relation to the Method, it is suggested that the semiquantitative method be explained in more detail. Explain in more detail from line 124-126.
In the inclusion criteria, articles up to April 2022 are considered, but from what year does the search start?
In Table 1, the evaluation column could be deleted since it is already explained in the text prior to the table. Consider whether this table is necessary since much of the information has already been presented previously.
In the Results section in the first lines (194-195) it would be interesting to mention the 38 articles that are located with other sources since, if not the data do not fit in this part, they are incomplete. In addition, it is recommended to improve the presentation of the tables and their content. For example:
Improve the format and content of Table 2. Perhaps only specify the categories or show only score and percentage. Improve the format of Table 3.
It is suggested that in Table 4 the corresponding authors be placed without numbering in the References section.
Try to ensure that the tables do not occupy more than one page and that they are not cut off.
The subtitle (3.4.4 Social and Cultural Variables) should be added at the top of page 14 and removed from the Note to Table 4.
In the Discussion section, it would be important to try to differentiate the findings according to the age groups of the children and adolescents. A 6-year-old child is not the same as a 16-year-old adolescent.
Specify the discussion to children and adolescents with ASD. In some parts of this section a very general discussion is made.
In the Conclusions section, a more concrete example of the idea provided in the last paragraph where future studies are proposed (multilevel, multilevel approach) could be added.
I encourage the authors to make the suggested changes to improve the quality of the work and the whole process.
Reviewer 2 Report
The authors performed an interesting review of the evidence on correlation of physical activity in children and adolescents with autism spectrum disorder (ASD) in low- or middle-income countries, using the McMaster Critical Review Form for Quantitative Studies.
The authors’ review concerns an important topic, aligned with the current research pointing to the fact that children and adolescents with ASD do not meet the international physical activity guidelines. It is an important public health concern that children and adolescents with ASD are more at risk of obesity than the general population at their age.
This study is a well-designed and structured review covering an interesting and topical research area.
Please see my comments and suggestions below:
- In the section ‘Methods’ (line 160), please move Table 1 to Supplementary Materials.
- In the section ‘Results (line 220), please move Table 2 to Supplementary Materials.
- In the section ‘Discussion’, please highlight to a greater extent the following themes: implications of this study and novelty in this study.
- In the section ‘Conclusions’, please describe the practical aspects of the review performed.
Reviewer 3 Report
The authors dedicated their paper to reviewing essential factors concerning the physical correlation between children and adolescents with an autism spectrum disorder. The prepared review was very thorough, nevertheless, the authors should include some additional data where their methodology is concerned. Therefore, please expand the definition used in this paper for "low- or middle-income countries"; It would also make the conclusion section more robust if the authors included more details as to what can be distinguished within "biological variables, social and cultural variables, and physical environment variables".
Also, the authors should underline the purpose of this research, i.e. maybe this material is going to be developed further and used in practice.
Reviewer 4 Report
Unfortunately, I donțt believe that this paper brings novelty to this field (ASD).
The correlations indicated by the authors are not specific to ASD. There are a lot of elements that could BIAS the results. The research is not useful in this format.
The authors should find other elements to convince that PA influences the incidence of ASD in children.
Round 2
Reviewer 3 Report
The authors have supplied responses to all of my questions. therefore I believe the paper can be accepted to be published.
Reviewer 4 Report
There is a considerable improvement in your work. The paper could still be improved but I appreciate that you did your best to give us a better version of the paper. Next time you should find a subject of debate that is easier to sustain. I think now you can publish your work, but I still suggest you be more thorough in your research.